# Exploring the Role of SGLT2 Inhibitors in Cancer: Mechanisms of Action and Therapeutic Opportunities

**DOI:** 10.3390/cancers17030466

**Published:** 2025-01-30

**Authors:** Aparamita Pandey, Martín Alcaraz, Pasquale Saggese, Adriana Soto, Estefany Gomez, Shreya Jaldu, Jane Yanagawa, Claudio Scafoglio

**Affiliations:** 1Division of Pulmonary and Critical Care Medicine, David Geffen School of Medicine, University of California Los Angeles, 700 Tiverton Drive, Los Angeles, CA 90095, USA; apandey@mednet.ucla.edu (A.P.); asoto07@g.ucla.edu (A.S.); stefgbruin24@g.ucla.edu (E.G.); shreyajaldu@g.ucla.edu (S.J.); 2Department of Biology and Biotechnologies Charles Darwin, University of Rome “Sapienza”, Piazzale Aldo Moro 5, 00185 Rome, Italy; pasquale.saggese@uniroma1.it; 3Department of Surgery, David Geffen School of Medicine, University of California Los Angeles, 700 Tiverton Drive, Los Angeles, CA 90095, USA; jyanagawa@mednet.ucla.edu

**Keywords:** SGLT2 inhibitor, cancer, glucose, metabolism

## Abstract

SGLT2 inhibitors, first introduced in the clinical practice for diabetes, have been suggested to be effective against different types of cancer. Here, we review the literature on the re-purposing of SGLT2 inhibitors for cancer, focusing on the data that are most relevant for human cancer.

## 1. Introduction

Cancer cells have an increased requirement of glucose compared with their normal counterparts. The pioneering studies by Otto Warburg showed that cancer is associated with a switch from oxidative phosphorylation to glycolysis, even in the presence of oxygen [1]. This was initially ascribed to a mitochondrial dysfunction, but subsequent studies have shown that cancer cells divert glucose utilization from complete oxidation to lactate production even in the presence of functional mitochondria [2]. This allows the fast production of energy through glycolysis, as well as the accumulation of glycolytic intermediates that can fuel biosynthetic pathways, providing building blocks (nucleotides, lipids, amino acids, hexosamines) for growing cells.

Glucose can be transported inside cancer cells through two independent mechanisms: (1) the canonical glucose transporters of the GLUT family, which are well known to be over-expressed in cancer [3]; (2) the sodium-glucose transporter (SGLT) family, whose role in cancer has started to be recognized more recently [2]. GLUTs are facilitative diffusion transporters, which rely on the gradient of glucose concentration across the plasma membrane [4]. There are 14 members of the GLUT family, each one with a specific pattern of physiological expression in normal tissues; GLUT1 and GLUT3 are the major transporters of this class found in cancer [5]. The expression of GLUTs in advanced cancer allows the in vivo measurement of their activity via 2-[^18^F] fluoro-deoxy-glucose (FDG) positron emission tomography (PET), commonly used for cancer staging. SGLTs are co-transporters that rely on the concentration gradient of Na across the plasma membrane to transport glucose. SGLTs are physiologically expressed in tissues specialized for active sugar transport: the intestine (SGLT1) and the kidney (SGLT1 and SGLT2) [6]. In the kidney, SGLT2 is expressed in the S1 segment and SGLT1 in the S2 segment of the proximal tubule; SGLT2, which is a high-capacity, low-affinity transporter, is responsible for re-absorption of 90% of filtered glucose, whereas SGLT1, which is a low-capacity, high-affinity transporter, is responsible for the remaining 10% [7]. Major differences between SGLT2 and GLUT1 are summarized in the Graphical Abstract.

Because of its crucial role in glucose reabsorption in the kidney, SGLT2 has been identified as a therapeutic target for diabetes, as inhibition of glucose re-absorption leads to loss of glucose in the urine, which lowers blood glucose [8]. Up to now, six different SGLT2 inhibitors (SGLT2i) have been approved by the Food and Drug Administration (FDA) for the treatment of diabetes: dapagliflozin, empagliflozin, canagliflozin, ertugliflozin, sotagliflozin, and bexagliflozin. Sotagliflozin and canagliflozin are dual SGLT1 and SGLT2 inhibitors. Recently, SGLT2is have been confirmed to have positive effects in patients with heart failure and chronic kidney disease, independent of the anti-diabetic effect. Pre-clinical evidence suggests that SGLT2is could have also an anti-cancer effect. However, the mechanism of anti-cancer action is not clear. Here, we review the current evidence on the role of SGLT2 in cancer and the possible use of SGLT2is as anti-cancer agents, focusing on the four molecules that have been on the market for longer time: canagliflozin, dapagliflozin, empagliflozin, and ertugliflozin.

## 2. SGLT2 Expression in Cancer

In lung cancer, the first report of SGLT2 expression was published by Ishikawa et al. in 2001 [9], who investigated the expression of SGLT1 and SGLT2 by RT-PCR, and observed an increased expression of SGLT2 in lymph nodes and liver metastasis compared with the primary tumor, but no difference between the primary tumor and adjacent lung tissue [9]. Subsequent studies in lung cancer have yielded inconsistent results. Zhang et al. reported an increased expression of SGLT2 in lung cancer compared with adjacent normal tissue, and a worse survival associated with SGLT2 expression [10]. Taira et al. observed the expression of SGLT2 by immunohistochemistry in mediastinal lymph-nodes that resulted in a false negative with the PET tracer FDG [11], consistent with the previously described transport activity of SGLTs, which have a low affinity for 2-deoxyglucose, but higher transport activity for 4-deoxyglucose [12]. These inconsistencies may be due to a lack of correlation between mRNA and protein level [13], to different affinities and specificities of the antibodies used, or to the different characteristics of patient populations and tumor types investigated in different studies. Non-small cell lung cancer is a heterogeneous group of diseases, encompassing two of the most frequent types, adenocarcinoma and squamous cell carcinoma, and many less frequent histological types. Within adenocarcinoma, there is a striking inter- and intra-tumor heterogeneity of morphology and differentiation state. We have observed the expression of SGLT2 in adenocarcinoma, with a significant correlation between SGLT2 expression and tumor grade: SGLT2 is over-expressed in well-differentiated and moderately differentiated lung adenocarcinoma but has reduced expression in poorly differentiated cancers [14]. Interestingly, SGLT2 up-regulation is an early event in lung carcinogenesis, with over-expression already present in pre-malignant lesions of the lung adenocarcinoma spectrum [14]. Conversely, TCGA analysis showed no difference in SGLT2 expression between cancer and normal adjacent tissue in lung squamous cell carcinoma [15,16]. A recent study reported expression of SGLT2 present more frequently in advanced-stage, aggressive adenocarcinoma patients [17].

In pancreatic cancer, two independent groups reported the increased expression of SGLT2 in cancer cells compared with adjacent normal tissue [18,19]. We confirmed the expression of SGLT2 in pancreatic and prostate adenocarcinomas, also measuring their functional activity in fresh tumor tissue by in vitro uptake of the SGLT-specific PET tracer methyl-4-[^18^F] fluoro-deoxy-glucose (Me4FDG) [20]. SGLT2 expression was also confirmed in breast cancer [21], kidney cancer [22], osteosarcoma [15], papillary thyroid cancer [23] and glioblastoma [24]. The functionality of SGLT2 in astrocytomas was confirmed in vivo by PET imaging in patients [24,25].

In conclusion, SGTL2 is expressed in several cancers, of both epithelial and non-epithelial origin, even if the data reported in the literature are, in some cases, inconsistent. Wu et al. observed a discordance between RNA and protein data: SGLT2 expression was not upregulated at the mRNA level in osteosarcoma. However, SGLT2 protein was overexpressed in osteosarcoma when compared to adjacent normal tissue and normal bone tissue [15]. We have also observed a discrepancy between mRNA level and protein expression. The discrepancies in immunohistochemistry between different reports can be due to the different antibodies used. Appendix A summarizes the studies that have investigated SGLT2 expression in cancer.

The human cancers in which SGLT2 expression has been demonstrated are summarized in Figure 1. The expression of SGLT2 in various cancers suggests a potential involvement of SGLT2 in neoplastic transformation. Since specific blockers of SGLT2-mediated glucose uptake are FDA-approved and currently used for diabetes, there is an interest in repurposing these drugs for cancer.

## 3. Pharmacology of SGLT2 Inhibitors

SGLT2is are a class of drugs that specifically inhibit glucose transport by SGLT2. Given the role of SGLT2 in glucose re-absorption in the kidney, these drugs have been approved by the FDA for the treatment of diabetes, and, more recently, for heart failure and chronic kidney disease.

### 3.1. Mechanism of Action

Since SGLT2 expression is physiologically restricted to the kidney tubules in mice and humans [6,26], SGLT2is are very selective in blocking glucose re-absorption in the kidney. Kidney reabsorption of glucose is characterized by a threshold due to the saturation of SGLTs in the tubule; normally, all filtered glucose is reabsorbed, but if the blood concentration of glucose exceeds 200 mg/dL, there is a linear relationship between blood glucose concentration and urinary glucose excretion. Glucose clamp experiments have shown that SGLT2is cause a left shift in the inflexion of the glucose threshold, inducing glycosuria at blood glucose concentrations as low as 80 mg/dL [27]. As a consequence, they lower blood glucose in diabetic patients [28,29] and improve insulin sensitivity with reduced circulating insulin levels [30]. Compensatory hepatic gluconeogenesis, occurring both in normal subjects and diabetic patients, prevents the development of hypoglycemia [31,32]. Bonner et al. showed that SGLT2 is expressed in pancreatic alpha cells, and SGLT2 inhibition in these cells stimulated the secretion of glucagon, which, in turn, triggers gluconeogenesis in the liver [33].

In addition to the glycemic control, SGLT2is have a plethora of beneficial systemic effects, even in non-diabetic patients [34]. The loss of calories caused by glycosuria brings about weight loss and a reduction in visceral fat [35], which is beneficial for cardiovascular health, and triggers a starvation-like response with production of ketone bodies [36]. SGLT2is also induce natriuresis, reducing blood pressure [37]. The increased delivery of sodium to the macula densa triggers the tubulo-glomerular feedback, causing constriction of the glomerular afferent arteriole and attenuating the increase in glomerular filtration rate that occurs in diabetic kidney disease [34]. The metabolic and hemodynamic effects of SGLT2is result in cardiovascular and renal protection independent of glycemic control [38,39]. These findings led to the FDA approval of SGLT2is for the treatment of heart failure and chronic kidney disease.

A widely recognized effect of SGLT2is that can be relevant for cancer is the anti-oxidant and anti-inflammatory action. A reduction in systemic inflammation can be due to the reduction of adipose tissue mass and reduced insulin levels. SGLT2is have been suggested to have direct effects on inflammation by reducing reactive oxygen species and the reduced secretion of inflammatory cytokines by macrophages [40].

### 3.2. Pharmacokinetics

After oral administration, SGLT2is are quickly absorbed with a bioavailability between 25% and 100% [41,42,43,44,45], a time to peak plasma concentration (Tmax) of 1–1.5 h [46,47], and a half-life between 5 and 20 h [43,47,48,49,50,51,52], allowing once-daily administration. The pharmacokinetic properties of the FDA-approved SGLT2is, at the doses approved for diabetes, are summarized in Appendix A. For the critical evaluation of pre-clinical results in the literature, it is important to note that SGLT2is have high binding to plasma protein: 86–99% [53,54,55,56], which limits the concentration available to diffuse in tissues and act on cancer cells. In addition, all these drugs have a high steady state volume of distribution, ranging from 73.8 to 9000 L [42,44,47,52,56,57], greater than the estimated human plasma volume, indicating extravascular distribution. In mice, a single dose of dapagliflozin, empagliflozin, or canagliflozin results in the accumulation of the drugs in the kidney and liver, with concentrations in these two organs 7–10 and 3–7 times higher than in plasma, respectively [46]. Autoradiography studies after a single intravenous administration of a single dose of radioactively labelled fluoro-dapagliflozin showed the sequestration of the drug in the lumen of kidney tubules [58]. This distribution further limits the amount of drug that is available for blocking SGLT2 in cancer cells. SGLT2is are also extensively metabolized and inactivated in the liver, mostly by glucuronidation [48,49,54,59].

Based on the reported pharmacokinetic properties, we estimate that the active concentration of the drugs that cancer cells can be exposed to in vivo ranges between 0.1 and 1 µm. These considerations are relevant as we move to summarize the evidence for the effects of SGLT2i on cancer.

## 4. Effect of SGLT2 Inhibitors on Cancer: Epidemiological and Clinical Evidence

In various epidemiological studies around the globe, researchers have explored the association between SGLT2is and cancer risk. Most of these retrospective studies have leveraged data from previous clinical trials or population-based data of SGLT2is in diabetic patients.

### 4.1. Initial Worrying Reports: Bladder and Breast Cancers

In 2011 the Food and Drug Administration (FDA) advisory committee voted against approving dapagliflozin for the treatment of diabetes, over safety concerns of increased bladder and breast cancer risk [60]. Since then, these concerns have been mostly refused and dapagliflozin is currently approved and used by patients all over the world.

A potential effect of SGLT2is on breast cancer risk has not been confirmed [61,62,63,64]. Some studies even found a reduced risk of breast cancer in diabetic patients treated with SGLT2i vs. DPP-4 inhibitors (HR: 0.51, *p* < 0.001) [65] or vs. non-SGLT2i users (HR: 0.77, *p* = 0.001) [66], and reduced breast cancer mortality in SGLT2i users vs. non-users [67].

For bladder cancer, the data are more controversial. Ptaszynska et al. (2015) conducted a pooled analysis of 21 clinical trials and did not confirm an increased risk of bladder cancer. The authors suggested that the previous report of higher incidence of bladder cancers in the dapagliflozin group compared with placebo was due to pre-existing bladder cancer, since 8 of 9 patients who developed bladder cancer during the study had microscopic hematuria at the time of recruitment [68]. Kohler et al. (2017) pooled data from 15 randomized trials with placebo, and two different empagliflozin doses (10 and 25 mg/d), and found no association between SGLT2i and the risk of bladder, renal, or breast cancer [62]. Other studies, like that by Dicembrini et al. (2019), also found no significant associations between SGLT2i and various types of cancer, including bladder cancer [69]. Spiazzi et al. (2023) reported no association with malignancies and no effect of SGLT2i on cancer-related survival, and concluded that SGLT2is do not increase the risk of bladder, renal, or breast cancer [70]. Abrahami et al. (2022) found no significant cancer risk associated with SGLT2i compared to GLP-1 receptor agonists or DPP-4 inhibitors in their international, multisite cohort study using the UK Clinical Practice Data Mart Database [71]. However, Pelletier et al. (2020) reported an association of bladder cancer with SGLT2is (OR 3.87), and specifically an increased risk of bladder cancer with empagliflozin (OR 4.49 vs. placebo and active comparators) [72]. Tang et al. (2017) performed a meta-analysis of clinical trials with SGLT2is in diabetic patients, and found that empagliflozin, but not canagliflozin and dapagliflozin, increased the risk of bladder cancer (OR 3.87, 95% CI 1.48–10.08). The effect was more pronounced in the trials with duration ≥52 weeks, and in older and overweight subjects [63]. The authors commented that diabetes and obesity are, in turn, risk factors for bladder cancer, and increased glucose concentration in the urinary tract and the increased frequency of urinary tract infections may contribute to an increased bladder cancer risk [63]. However, two different population-based studies from Hungary and from Taiwan found a reduced bladder cancer risk in diabetic patients treated with SGLT2i vs. those treated with DPP-4 inhibitors [73,74] and vs. non-SGLT2i users [66]. Overall, the safety of SGLT2is for bladder cancers needs to be confirmed by larger prospective studies. The presence of several confounding factors, including obesity, diabetes, and concomitant medications, as well as the heterogeneity of the patient populations included in the studies and the short follow-up, complicates the interpretation of the epidemiological data available. The increasing utilization of SGLT2is in clinical trials in non-diabetic patients and the continued monitoring of study subjects for longer follow-up time will likely provide clearer answers.

### 4.2. Effect of SGLT2is on Overall Cancer Risk

Recent epidemiological studies suggest that SGLT2i treatment may offer protection against various cancer types. When SGLT2is are analyzed as a group, the studies tend to show improved outcomes. Chung et al. (2022) found SGLT2i use to be associated with reduced cancer mortality (HR: 0.73) after adjusting for age, gender, comorbidities, and concurrent medications [75]. Results were confirmed by Hu et al. [66]. Similarly, Sung et al. found reduced incidence of cancer in patients using SGLT2i vs. DPP-4 inhibitors (HR: 0.90, 95% CI: 0.87–0.93) [74]. Chung et al. (2023) concluded that the use of SGLT2i is associated with reduced overall cancer-related mortality (HR: 0.58, *p* < 0.001) and reduced incidence of cancer compared with DPP-4 inhibitors (HR: 0.7, *p* < 0.001) [65]. Huang et al. (2024) found reduced cancer-specific mortality (HR: 0.21, *p* < 0.001) and improved 5-year cancer-specific overall survival (89.6% vs. 63.7%) in SGLT2i-users vs. non-users. An analysis of interaction terms showed that the benefit of SGLT2is decreases with age, cancer stage, and severity of diabetes [67]. Spiazzi et al. (2023) found that SGLT2i treatment did not increase or reduce the risk of cancer nor overall cancer mortality [70]. Interestingly, Perelman et al. reported improved survival in patients treated with SGLT2i and immune checkpoint inhibitors (ICI) vs. diabetic patients treated only with ICI [76].

When the effects of single SGLT2is are considered separately, the results are less clear. Shi et al. (2021) reported mixed findings: compared with placebo, SGLT2i did not increase the risk of cancer, except for ertugliflozin (RR 1.80, 95% CI: 1.02, 3.17, *p* = 0.04). Compared with other hypoglycemic drugs, dapagliflozin increased the risk (RR: 2.71, 95% CI: 1.14–6.43, *p* = 0.02), whereas empagliflozin reduced the risk (RR: 0.67, 95% CI: 0.45–0.98, *p* = 0.04). Compared with placebo, only empagliflozin increased the overall risk of cancer (RR: 1.25, 95% CI: 1.05-1.49, *p* = 0.014) [64]. Conversely, Benedetti et al. (2022) observed reduced overall cancer risk with SGLT2is (RR = 0.35, CI 0.33–0.37, *p* = 0.00). When they analyzed the effect of individual inhibitors, dapagliflozin and ertugliflozin were associated with lower cancer risk (RR = 0.06, CI 0.06–0.07 and RR = 0.22, CI 0.18–0.26), and empagliflozin with higher risk (RR = 1.54, CI 1.17–1.02) [77]. Conversely, Sung et al. found that dapagliflozin and empagliflozin reduced the risk of cancer compared with DPP-4 inhibitors, whereas canagliflozin did not have an effect [74].

These contrasting results can be attributed to the heterogeneity of the studies and of the control groups, with different patient populations, comorbidities, and concomitant medications, as well as a possible cancer-specific effect of SGLT2is that complicates the interpretation of overall cancer incidence and mortality studies. 

### 4.3. Effect of SGLT2i on Specific Cancers

In the studies that were able to examine the effect of SGLT2is on specific cancer types, the most convincing evidence is available for non-small cell lung cancer and gastrointestinal cancers (colorectal cancer, hepatocellular carcinoma, pancreatic cancer).

For non-small cell lung cancer, Sung et al. reported a reduced risk in SGLT2i users vs. DPP-4 inhibitor users (HR = 0.87, 95% CI = 0.80–0.95) [63], and Huang et al. reported a significant reduction in mortality with SGLT2i use (HR: 0.26, *p* < 0.001) [67]. Luo et al. found that SGLT2is were associated with significantly reduced overall mortality (HR = 0.68, 95% CI = 0.60–0.77) and lung cancer-specific mortality (HR = 0.69, 95% CI = 0.58–0.83), with stronger effects associated with longer duration of use (HR = 0.54, 95% CI = 0.44–0.68) and the concomitant use of SGLT2i and metformin (HR = 0.62, 95% CI = 0.54–0.72) [78].

Some studies have reported the effect of SGLT2is on gastrointestinal cancers without categorizing by the organ of origin, and concluded that canagliflozin reduced the risk [63,72], whereas empagliflozin increased the risk [64]. For colorectal cancer, reduced incidence has been reported in patients treated with SGLT2is vs. those treated with DPP-4 inhibitors [74], especially in male patients and in patients < 65 years old [79]. Huang et al. reported reduced mortality (HR: 0.18, *p* < 0.001) [67], and Chiang et al. reported increased 5-year overall survival (OS: 86.2% vs. 62.3%, *p* = 0.013) and progression-free survival (PFS: 76.6% vs. 57.0%, *p* = 0.021) in diabetic patients treated with SGLT2i vs. non-SGLT2i recipients [80]. For hepatocellular carcinoma, Sung et al. reported a reduced risk in SGLT2i users vs. DPP-4 inhibitor users (HR = 0.90, 95% CI = 0.84–0.98) [74], Huang et al. reported reduced mortality (HR: 0.19, *p* < 0.001) [67] and Hendryx et al. found that SGLT2i use is associated with significantly reduced mortality (HR: 0.68), with a more pronounced effect with longer duration of exposure [81]. For pancreatic cancer, Tanaka et al. found that the cumulative administration of at least 180 days of SGLT2i significantly reduced the incidence of pancreatic cancer [82]. Sung et al. reported a reduced risk in SGLT2i users vs. DPP-4 inhibitor users (HR = 0.84, 95% CI = 0.73–0.97) [63], and Huang et al. reported reduced mortality in SGLT2i users (HR: 0.46, *p* < 0.001) [67].

Despite the reported expression of SGLT2 in prostate cancer, Cui et al. found no change in prostate cancer incidence in diabetic SGLT2i users in a meta-analysis of 47 studies [83]. Tang et al. also reported no difference in incidence [63].

Appendix A summarizes the methods, study durations, and databases utilized for these epidemiological studies. The conflicting and inconsistent results could be due to the existing confounding factors, including the type and doses of other anti-diabetic drugs (metformin, secretagogues, insulin, DPP-4 inhibitors, GLP1 agonists), the degree of glyco-metabolic control, other commonly used drugs (anti-inflammatory drugs, statins), and the severity of diabetic disease. Therefore, the clinical evidence coming from retrospective studies needs to be confirmed by randomized placebo-controlled prospective trials.

### 4.4. Clinical Trials

To date, there has been only one published clinical trial of SGLT2is in cancer. Park et al. [84] performed a phase Ib study in patients with metastatic or locally advanced pancreatic ductal adenocarcinoma, in addition to standard of care gemcitabine and nab-Paclitaxel (GnP). The addition of 2 months of oral dapagliflozin (10 mg/day) to GnP was well-tolerated. Of the 12 evaluable patients, 2 had a partial response (PR), 9 had stable disease (SD), and only one had progressive disease (PD). So, 91% of patients had either SD or PR; other studies have reported a rate of disease control in patients on GnP alone of 48% [85]. The authors concluded that there were potential signs of dapagliflozin’s efficacy as a treatment for PDAC when added to standard GnP therapy [84].

Epidemiological and clinical studies suggest that SGLT2is may have a role in cancer treatment. The mechanisms of action can be more deeply investigated in pre-clinical models. In the following sections, we review the pre-clinical, in vivo and in vitro evidence for the efficacy of SGLT2is against cancer. The mechanisms of the anti-cancer action of SGLT2is are summarized in Figure 2 and Figure 3. 

## 5. SGLT2 Inhibitors as Anti-Cancer Agents: In Vivo Preclinical Evidence

The clinical relevance of SGLT2is as putative anti-cancer agents has been evaluated in several preclinical trials. A summary of in vivo studies is presented in Appendix A. The inhibitory effect of gliflozins has been shown in different types of tumor, including lung, breast, and hepatocellular carcinoma.

### 5.1. Lung Cancer

Scafoglio et al. demonstrated that early-stage lung adenocarcinoma (LUAD) relies on SGLT2 activity for glucose uptake. Canagliflozin treatment significantly reduced tumor size and improved survival in mice compared with the placebo group [14]. Despite a consistent and significant reduction in tumor burden in a Kras^G12D^-driven genetically engineered tumor model and in patient-derived xenografts, SGLT2 inhibition was not able to completely block tumor development [14], and the tumors that developed during chronic canagliflozin treatment were less differentiated than tumors in the placebo group [86].

Saggese et al. confirmed the positive effect of SGLT2i on LUAD tumor burden with a different SGLT2i, empagliflozin, and characterized in detail the molecular mechanism underlying glucose starvation-induced de-differentiation: glucose deprivation causes the depletion of alpha-ketoglutarate and unbalanced activity of histone methyl-transferase EZH2 due to insufficient activity of alpha-ketoglutarate-dependent histone demethylases [87], ultimately leading to tumor de-differentiation [86].

An effect of canagliflozin on lung cancer was confirmed in tumor xenografts with different cell lines (A549, H1299, and H1975) in nude mice, where canagliflozin did not have an effect as a single agent but potentiated radiotherapy, an effect associated with AMPK inhibition [88] (Figure 2).

The expression of SGLT2 in murine (and human) lung adenocarcinomas seems to suggest a direct inhibition of SGLT2is on cancer cell glucose uptake, as confirmed by PET imaging studies with the SGLT-specific tracer Me4FDG [14]. However, Hsieh et al. showed that canagliflozin can be also effective in lung squamous cell carcinomas, which do not express SGLT2 [16], confirming that a systemic effect of SGLT2is also plays a role in their anti-cancer effect (Figure 3).

### 5.2. Breast Cancer

SGLT2is can inhibit in vivo the growth of estrogen-dependent [21] and triple-negative BC cells [89]. The effect on triple-negative breast cancer has been suggested to be systemic, by reducing circulating insulin levels [90,91].

### 5.3. Hepatocellular Carcinoma

SGLT2is have been tested in different xenograft tumor models of hepatocellular carcinoma (HCC). Canagliflozin suppressed tumor growth and prolonged survival in mice carrying HCC xenograft tumors by blocking β-catenin activation [92] (Figure 2), by reducing cell proliferation and neo-angiogenesis [93], and potentiated doxorubicin [94]. The effect was not present in an SGLT2-negative cell line, suggesting a direct effect of SGLT2i on the tumor [93].

SGLT2is have been proposed to attenuate hepatocarcinogenesis in different models of metabolic dysfunction-associated steatohepatitis (MASH) [95,96]. Canagliflozin reduced liver steatosis and improved the metabolic profile of animals, even when the tumors did not express SGLT2, so the beneficial effect of canagliflozin was attributed to reduced inflammation in the liver and in the adipose tissue [95] (Figure 3). Interestingly, empagliflozin in combination with metformin significantly improved survival, and reduced necroinflammatory changes, liver volumes, and cell proliferation in a carcinogen-induced model of HCC [97].

### 5.4. Other Cancers

The efficacy of SGLT2is in pancreatic cancer has only been assessed in xenograft models [19,20,98].

In prostate cancer, canagliflozin slowed tumor growth and improved mice’s survival in an androgen-independent xenograft model, both as single agent and additively with radiotherapy (5 Gy) [99].

In colorectal cancer, surprisingly Korfhage et al. observed a sex-dependent deleterious effect of a canagliflozin-containing chow diet in a model of intestinal adenomatous polyposis. The authors speculated that the high intraluminal concentration of canagliflozin in the intestine leads to the inhibition of SGLT1, which increases the luminal glucose available for adenomas to grow [100]. These observations are not consistent with the beneficial effect observed in epidemiological studies.

In osteosarcoma, canagliflozin significantly reduced tumor growth in a subcutaneous xenograft model, and the effect was additive with the activator of the STING pathway 2′3′-cGAMP [15]. The STING pathway, which was up-regulated in vitro by SGLT2 inhibition [15], is an inducer of anti-cancer immunity through activation of type I interferons [101].

SGLT2is were also effective in xenograft models of glioblastoma [102], papillary thyroid carcinoma [23], renal cell carcinoma [103], and cervical cancer [104].

## 6. SGLT2 Inhibitors as Anti-Cancer Agents: In Vitro Evidence

Many publications have investigated the efficacy of SGLT2 in vitro. Villani et al. screened a panel of cell lines from different cancers and found out that canagliflozin, but not dapagliflozin, can successfully reduce the proliferation and clonogenic survival, but not viability, of prostate, lung, liver, and breast cells, but not of colon and ovarian cells [105]. This is accompanied by AMPK activation, as well as inhibition of mitochondrial complex I [105] (Figure 2). In lung, colon, prostate, and pancreatic cancer, canagliflozin, dapagliflozin and ipragliflozin reduce proliferation and synergize with the chemotherapeutic drug doxorubicin [106].

In lung cancer, Yamamoto et al. reported that canagliflozin has an anti-proliferative effect in vitro but not in vivo [107], suggesting an off-target effect. Canagliflozin has been suggested to inhibit mutant EGFR independently of SGLT2 [108].

In breast cancer, canagliflozin and dapagliflozin cause cell cycle block [109] and reduce viability [106], possibly by targeting the AMPK/mTOR pathway [21] (Figure 2). Canagliflozin, but not dapagliflozin, reduces proliferation and mitochondrial respiration, independently of glucose concentration and SGLT2 expression: the authors suggest that this effect is due to the inhibition of glutamate dehydrogenase, with a blockade of glutamine anaplerosis [109]. Nasiri et al. showed that, while dapagliflozin is very effective in vivo by reversing hyperinsulinemia in obese mice, it has no effect in vitro at pharmacologically achievable concentrations [90]. Eliaa et al. showed that canagliflozin potentiates the anti-breast cancer action of doxorubicin, suggesting that this combination may also reduce the cardiotoxicity associated with doxorubicin treatment, given the known cardioprotective effect of SGLT2is [110].

In pancreatic cancer, canagliflozin and sotagliflozin (both dual SGLT1/2 inhibitors) reduce cell proliferation and migration, and down-regulate Yap1 signaling [19] (Figure 2). In this publication, the authors use a concentration of canagliflozin (1 μM) that is clinically achievable, and confirm the results with an siRNA targeting SGLT2. Xu et al. showed that canagliflozin reduces cell proliferation and induces apoptosis in pancreatic cancer. However, the IC50 for cell viability inhibition is around 60 μM, around 10,000 higher than the IC50 measured for SGLT-dependent glucose uptake. This suggests that the observed effect is off-target [98].

In hepatocellular carcinoma, the mechanism of action is unclear. Nakano et al. report that canagliflozin blocks hepatocellular carcinoma (HCC) proliferation by downregulating the mitochondrial electron transport chain and activating the AMPK pathway [111]. Luo et al. confirmed that canagliflozin reduces cell viability and migration in different HCC and hepatoblastoma lines, and this effect was associated with the reduced activation of the Akt pathway and reduced HIF-1α stabilization [112] (Figure 2). Kaji et al. showed that canagliflozin reduces cell proliferation and induces apoptosis in Huh7 and HepG2 cells, which express SGLT2, but not in HLE cells, which do not have SGLT2. The dose used is still very high, 10 μM [93], and reduces the uptake of 2-deoxy-glucose, which has been reported to be a GLUT, not an SGLT, substrate [12,113]. Hung et al. suggested that canagliflozin blocks GLUT-dependent glucose transport in cancer cells, thus inhibiting β-catenin signaling [92]. SGLT2is have been used as combination treatments in liver cancer. Dapagliflozin reduced cisplatin resistance in hepatoblastoma cells [114], and canagliflozin sensitized HepG2 cells to irradiation, concomitantly with the inhibition of the PI3K/AKT/mToR pathway and β-catenin [115] (Figure 2).

In glioblastoma, Shoda et al. reported that canagliflozin reduced cell viability, reduced glucose uptake, and activated AMPK. The effects on viability and AMPK activation were phenocopied by siRNA targeting SGLT2, excluding an off-target effect [102]. However, canagliflozin did not have an effect at 5, 10, or 20 µm, but only at 40 µm, a difficult concentration to achieve in vivo. At this concentration, canagliflozin inhibited the uptake of 2DG, suggesting an inhibitory effect on GLUTs [102].

In colorectal cancer, dapagliflozin reduced the cell number and viability [116,117].

In renal cell carcinoma, dapagliflozin reduced cell viability and cell cycle, and increased apoptosis [103].

Moreover, canagliflozin was found to be effective at suppressing prostate cancer growth through the inhibition of glycolysis through the PI3K/AKT/mTOR pathway, inhibiting HIF-1a, which is a transcription factor for GLUT-1 and LDHA [99] (Figure 3).

Molecular mechanisms of SGLT2is in vitro have been amply described in several review articles [118,119,120]. Most in vitro papers used supra-pharmacological concentrations of SGLT2is and the presented mechanisms may or may not be relevant for human cancer treatment. In Appendix A we listed details of these studies and mentioned the possible off-target effects. Figure 2 summarizes the molecular mechanisms of SGLT2is on cancer cells.

The in vitro and in vivo studies show that SGLT2is have a complex mix of direct effects on the tumor, driven by the blockade of glucose transport in cancer cells, and of indirect effects, due to systemic metabolic adaptations to the treatment. In the next section, we review the mechanisms of systemic and microenvironmental changes induced by SGLT2is that can affect cancer.

## 7. Mechanisms of Anti-Cancer Action of SGLT2 Inhibitors: Systemic and Tumor Microenvironmental Effects

Recent advances in cancer research have highlighted the nature of cancer as a systemic disease, driven by complex interactions of genetic and environmental factors, and the crosstalk of multiple systems in the body [121]. SGLT2is exert a variety of systemic effects that can influence cancer progression through multiple metabolic, hormonal, and inflammatory pathways.

### 7.1. Hormonal Effects

One of the main effects of SGLT2is is the reduction in circulating insulin [30] (Figure 3). In addition to being a regulator of glucose metabolism, insulin is an anabolic hormone involved in the promotion of cancer progression [122] by activating the PI3K/AKT pathway [123]. It is likely that a reduction in insulin levels contributes to the anti-cancer effects of insulin, especially on cancer types that have been linked with obesity, metabolic syndrome, and hyperinsulinemia, including gastrointestinal and breast cancers [124].

SGTL2is are also effective at reducing leptin and increasing adiponectin in diabetic patients [125,126,127,128,129] (Figure 3). Leptin is an anorexigenic hormone produced by the adipose tissue proportionately to the expansion of the fat mass [130], and activates the JAK/STAT, PI3K/AKT, and MAPK pathways in target cells [131]. Leptin is considered to promote cancer by directly stimulating cancer cells and by modulating anti-cancer immune responses [132]. Adiponectin is a cytokine also produced by adipose cells, which improves insulin sensitivity, reduces inflammation, and its circulating levels are reduced in cancer patients [133]. Adiponectin can also have a direct effect on cancer cells through AMPK activation [133]. The reduction of leptin and stimulation of adiponectin can contribute to the anti-cancer effects of SGLT2is.

### 7.2. Ketone Bodies

A key systemic effect of SGLT2is is the elevation of serum β-hydroxybutyrate (BHB), a ketone body with anti-inflammatory and antioxidative properties [134]. SGLT2is, by causing glucose loss in the urine, induce a starvation-like response in the liver, with increased BHB production (Figure 3). Ketone bodies can have a beneficial effect against cancer via different mechanisms. BHB can activate the cell surface G-coupled receptor HCAR2 (or GPR109A) [135] or can be imported in cancer cells via monocarboxylate transporters (MCT) 1 and 2 [136], frequently over-expressed in cancer [137]. HCAR2 activation is required for the activation of Hopx, a key regulator of cell differentiation with ubiquitous expression in human tissues [138], to hinder colorectal cancer development [139]. HCAR2 is expressed mainly in adipose tissue and immune cells, including macrophages [140] and neutrophils [141], and its activation reduces obesity-associated adipose tissue inflammation [142]. BHB is also a known epigenetic regulator, inhibiting the deacetylation of histone H3 on lysine 9 and 14 [143], and can be directly conjugated to lysine residues on histone tails (β-hydroxybutyrylation) [144] and on other proteins, modulating cellular metabolism [145,146]. Finally, ketone bodies can directly scavenge reactive oxygen species (ROS) [147]. Therefore, ketone bodies are potentially important mediators of the effects of SGLT2is on cancer.

### 7.3. Inflammation and Immune Modulation

Low-grade chronic inflammation has been linked with the development of several diseases, including diabetes, cardiovascular diseases, and cancer [148]. SGLT2is reduce inflammation, potentially modulating anti-cancer immune responses [149] (Figure 3). SGLT2 inhibition has been associated with reduced circulating levels of major regulators of systemic inflammation in diabetic patients: IL-6 [127,150], TNF-α [151], and C-reactive protein, a clinically relevant marker of systemic inflammation [128,129]. Mechanistically, the effect on systemic inflammation is likely dependent on a complex interplay of different effects and pathways.

SGTL2is reduce the adipose tissue mass [35,152], and inflammatory changes are induced by the expansion of the adipose tissue [153] (Figure 3). SGLT2is reduce liver steatosis and injury [154,155,156], an important driver of systemic inflammation in patients with metabolic disease [157]. In pre-clinical models, the anti-steatosis effect was accompanied by AMPK activation in the liver [158] (Figure 3). SGLT2is cause pseudo-starvation and AMPK activation in different tissues [159,160]. AMPK has been associated with the suppression of systemic inflammation [127,161]. The anti-inflammatory action of SGLT2is on pre-clinical models of inflammatory bowel disease and lung fibrosis have been associated with the activation of AMPK and the anti-oxidant Nrf2 pathway [162,163].

SGLT2is also improve endothelial function [164,165,166], possibly by reducing advanced glycation end-products (AGEs), a known mediator of diabetic microvascular dysfunction [167] (Figure 3). The reduction of AGEs likely plays a role in the anti-inflammatory and anti-oxidant effect of SGLT2is [168]. SGLT2is hinder oxidative stress, as measured by circulating 8-hydroxy-2′-deoxyguanosine in patients [126,169]. In pre-clinical models, SGLT2is have been shown to reduce free radical-producing NAPDH oxidase in the kidney [170,171] and endothelial cells [170,172] by inhibiting mitochondrial ROS production [173] and activating AMPK-dependent mitophagy in endothelial cells [174], and by reducing the production of pro-inflammatory 20-hydroxyeicosatetraenoic acid (20-HETE) in a diabetic kidney model [175].

As for the effects on specific immune cells, SGLT2is inhibit the NLRP3 inflammasome in macrophages, at least in part by increasing ketone bodies [40], and empagliflozin promotes the anti-inflammatory M2 macrophage polarization in white adipose tissue, reducing systemic inflammation and insulin resistance [176]. Empagliflozin inhibits pro-inflammatory Th17 and increases anti-inflammatory Treg differentiation of CD4^+^ T cells, via mTOR inhibition [177,178]. This effect is expected to reduce systemic inflammation and reduce cancer risk, but in the tumor microenvironment the induction of Tregs is known to suppress anti-cancer immune responses. The role of SGLT2is in the balance between Th17 and Tregs in the tumor microenvironment should be assessed in prospective studies. Canagliflozin suppress tumor growth in osteosarcoma by activating the STING-IRF3/IFN-β pathway and promoting immune cell infiltration. Combined treatment with SGLT2is and a STING agonist enhances these antitumor effects [15]. Finally, canagliflozin enhances the immune-mediated clearance of senescent cells through the AMPK-mediated inhibition of immune checkpoint receptor PD-L1, a mechanism that is also potentially relevant for cancer immunosurveillance [179].

Less explored anti-inflammatory mechanisms of SGLT2is are the reduction of serum uric acid [180], which causes oxidative stress and inflammation [181], via increased renal excretion [182], and the inhibition of microbiome-derived inflammatory uremic toxins [183] like p-cresol [184].

## 8. Conclusions

In conclusion, the epidemiological, clinical, and pre-clinical evidence suggests that SGTL2 may be a novel therapeutic target in different types of cancer. The evidence is more solid for lung, breast, liver, and pancreatic cancer. The effect of SGLT2is is likely very complex and involves both the direct inhibition of glucose uptake in cancer cells and systemic effects. In addition to class effects, there may be differences according to the molecule used, since some off-target effects like mitochondrial complex I inhibition by canagliflozin contribute to the anti-cancer action. The weight of cancer-specific vs. systemic effects may be different according to the cancer type: the systemic effects are likely more relevant for obesity-associated cancers, such as breast and gastrointestinal tumors. These findings warrant further investigation into SGLT2’s broader implications in cancer and its treatment. Overall, targeting SGLT2 is a promising strategy against cancer.

Clinical trials assessing the efficacy of SGLT2is in various cancer types will be essential to validate their therapeutic potential. A search on clinicaltrials.gov shows there are several ongoing prospective early-phase clinical trials on SGLT2is for cancer. Groups at Washington University, St. Louis are investigating the role of dapagliflozin as a neoadjuvant treatment for high-risk patients with prostate cancer (NCT04887935) and for pediatric brain tumors (NCT05521984). SGLT2is are being investigated to prevent hyperglycemia and improve the anti-cancer effects of PI3K inhibitors in PI3KA-mutant breast cancer (NCT05090358) and in various advanced solid tumors (NCT04073680), and in protecting against doxorubicin cardiotoxicity in breast cancer (NCT06341842, NCT06103279). These studies aim to leverage the metabolic and cardioprotective effects of SGLT2is to reduce toxicity, and potentially improve efficacy, of standard of care chemotherapy. Both the systemic and the tumor-direct effects of dapagliflozin are being investigated in a phase 1 study in hyperinsulinemic women with HER2-negative breast cancer undergoing neoadjuvant treatment (NCT05989347). These early clinical trials will lay the foundation for larger efficacy studies. An analysis of biospecimens from these studies will also allow the investigation of the relevance of mechanisms of action found in pre-clinical models for human cancer.

Investigating the molecular mechanisms underlying the anti-cancer effects of SGLT2is is crucial, particularly in distinguishing between systemic and local effects. SGLT2is have been shown to block glucose uptake in cancer cells, thus leading to AMPK activation. The AMPK pathway is a known regulator of cancer growth and progression, inhibiting several other pro-cancer pathways including mToR, HIF-1α, β-catenin, and Yap-1 (Figure 2). On the other side, SGLT2is are also known to exert systemic effects that are potentially beneficial against cancer, including reduced fat mass and the reduced secretion of insulin, which is a growth factor for cancer cells, reduced advanced glycation end products, and oxidative stress in endothelial cells, leading to reduced systemic inflammation, the reduced accumulation of fat in the liver, and the increased production of ketone bodies, which can also have anti-cancer properties (Figure 3). Further elucidation of the mechanisms that are relevant for human cancer is required, as well as the study of SGLT2i interactions with standard-of-care therapies such as immunotherapy and chemotherapy. A significant challenge in these trials is determining the optimal timing for initiating SGLT2i treatment—whether it should be introduced in the early or late stages of cancer. Our studies in lung adenocarcinoma suggest that SGLT2 is expressed in the earliest stages of cancer development, and pre-malignant lesions are exquisitely sensitive to SGLT2is because they do not express glucose transporters of the GLUT family. This observation suggests that SGLT2is will be most effective for the chemoprevention, rather than the treatment, of lung adenocarcinoma.

Exploring the molecular mechanisms by which SGLT2 may contribute to carcinogenesis will provide insights into its role in cancer biology. Furthermore, identifying biomarkers that predict treatment response and tumor adaptations can aid in optimizing therapeutic strategies.

## Figures and Tables

**Figure 1 cancers-17-00466-f001:**
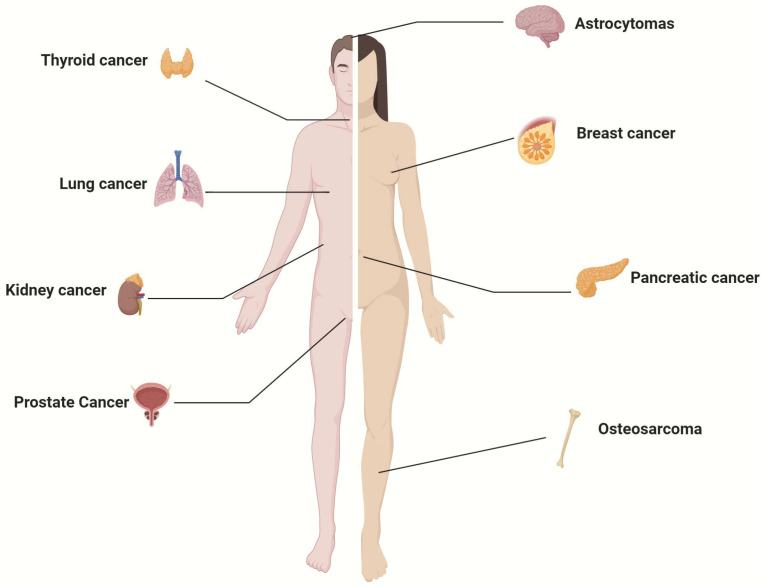
Human cancers known to express SGLT2.

**Figure 2 cancers-17-00466-f002:**
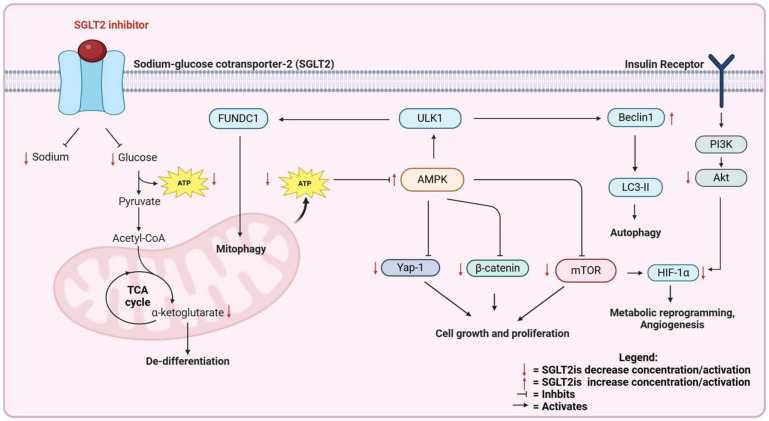
Direct effects of SGLT2 inhibitors on cancer cells.

**Figure 3 cancers-17-00466-f003:**
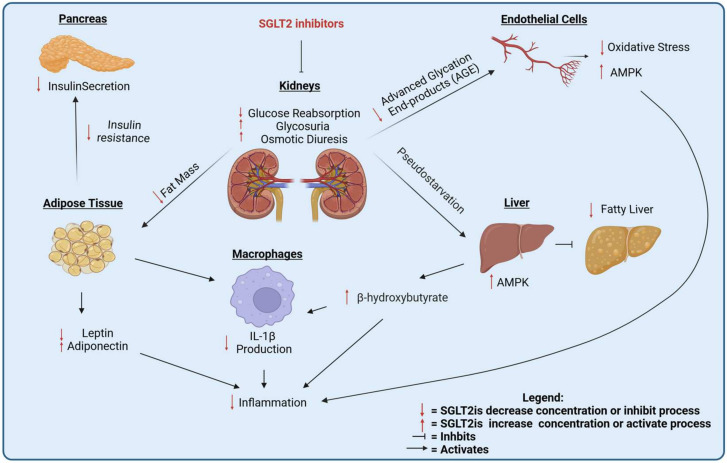
Systemic effects of SGLT2 inhibitors.

## Data Availability

No new data were created or analyzed in this study. Data sharing is not applicable to this article.

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
