# Peer review of "Exploring the Role of SGLT2 Inhibitors in Cancer: Mechanisms of Action and Therapeutic Opportunities"

_cancers, 2025, doi:10.3390/cancers17030466_

Round 1

Reviewer 1 Report

Comments and Suggestions for Authors

1)      In the abstract, the aim of the work should be added after line 16. Moreover, if the numerical results are available from previous works, I would recommend removing descriptive results and adding the numerical results.

2)      Adding a graphical abstract at the end of the introduction is appreciated. The authors could use Figure 1 as a graphical abstract if they add to the whole of the co-factors contributing to the transport system and indicate the inhibition places for therapeutic agents.

3)      Why did the authors select only lung cancer in section 2?

4)      Pharmacodynamics is related to drug-target interactions, in section 3.1. The authors did not investigate pharmacodynamics at all since they did not introduce a target, nor investigate binding energy and not molecular docking.

5)      The authors should write a table to list SGLT2 inhibitors along with pharmacokinetic (absorption, distribution, and metabolic) and pharmacodynamic properties.

6)      It is good for the authors to discuss SGLT2 inhibitors in healthy tissues.

7)      It is good to show in figures and discuss molecular mechanisms of anticancer effects of SGLT2 inhibitors (lines 627-640). The authors only wrote investigating the molecular mechanisms is crucial. Of course is important but what is their contribution? 

Author Response

Thank you for the insightful comments. We addressed all the issue raised, as follows.

1) In the abstract, the aim of the work should be added after line 16. Moreover, if the numerical results are available from previous works, I would recommend removing descriptive results and adding the numerical results.

We added a statement (highlighted in yellow) that the aim of our work is to perform a narrative review of the literature on SGLT2 inhibitors in cancer. The numerical data available in the literature would be too much to include in the abstract, since there are several publications with different outcomes, which are discussed in the body of the article.

2)      Adding a graphical abstract at the end of the introduction is appreciated. The authors could use Figure 1 as a graphical abstract if they add to the whole of the co-factors contributing to the transport system and indicate the inhibition places for therapeutic agents.

We changed Figure 1 as suggested and used it as graphical abstract at the end of the introduction.

3)      Why did the authors select only lung cancer in section 2?

We presented lung cancer first because there is the most evidence for SGLT2 expression for this kind of cancer. The other cancers were also presented in section 2.2. Since this subsection structure was confusing, we removed it and now we present all cancers in the same section.

4)      Pharmacodynamics is related to drug-target interactions, in section 3.1. The authors did not investigate pharmacodynamics at all since they did not introduce a target, nor investigate binding energy and not molecular docking.

We replaced "Pharmacodynamics" with "Mechanism of action". We think that the investigation of the molecular docking would be beyond the scope of this review, which is more focused on translational applications.

5)      The authors should write a table to list SGLT2 inhibitors along with pharmacokinetic (absorption, distribution, and metabolic) and pharmacodynamic properties.

Some of this information was already present in Table S2. We expanded this table to include all the requested information.

6)      It is good for the authors to discuss SGLT2 inhibitors in healthy tissues.

Thank you. The discussion of SGLT2 inhibitors in healthy tissues is in lines 140-153.

7)      It is good to show in figures and discuss molecular mechanisms of anticancer effects of SGLT2 inhibitors (lines 627-640). The authors only wrote investigating the molecular mechanisms is crucial. Of course is important but what is their contribution? 

Thank you. The molecular mechanisms are presented in figure 3 and 4 and in sections 6 and 7. We added a mention of these mechanisms in the conclusion, in lines 601-610, as requested.

Reviewer 2 Report

Comments and Suggestions for Authors

The Authors provide a thorough, and detailed overview on the potential role of SGLT2i in cancer incidence and prognosis, ranging from experimental to epidemiological data. It is an extensive and well-written review on an interesting topic. I have some minor comments and suggestions that the Authors should address.

 Specific issues

 1)               Clinical evidence and epidemiological data come from observational studies, or from clinical trial not designed to verify the effect of SGLT2i on cancer. Therefore, many confounding factors could have influenced the results. First, the use of other antidiabetic drugs potentially influencing cancer incidence and outcomes (eg, metformin, secretagogues, insulin, GLP1Ras…). Also, different types of insulin used, mean daily doses of insulin, and glyco-metabolic control could play a role, and some other commonly used drugs could also interfere for these outcomes (eg. ASA and statins). Even if I personally agree with the pathophisiologically-based speculation of a positive effect of SGLT2i on cancer, these factors should be highlighted as potential limitations when describing epidemiological and clinical evidence.

2)               Since experimental and epidemiological data are conflicting and sometimes disappointing, as correctly admitted by the Authors, I think that some statements in the manuscript are exaggeratedly enthusiastic and should be attenuated, eg:

a.       Abstract: “Epidemiological evidence in diabetic patients suggest that individuals treated with SGLT2 inhibitors have lower incidence and better outcomes of cancer.”

b.       Abstract: “In conclusion, there is substantial evidence that SGLT2 inhibitors are effective against different types of cancer.”

c.       Clinical Trials (Line 331): “Epidemiological and clinical studies provide evidence for the relevance of SGLT2is in cancer treatment.”

d.       Conclusion: “the epidemiological, clinical, and pre-clinical evidence confirmed that SGTL2 is a therapeutic target in different types of cancers”

3)               The manuscript is very detailed, and overall could result difficult to read. I would suggest shortening some paragraphs (eg, “in vivo preclinical evidence) by taking advantage of the useful tables provided as supplements.

 Minor issues

 1)               SGLT2 inhibitors are sometimes indicated extensively, sometimes as SGLT2is. Please, unify.

2)               Lines 204, 249, 252, 278: “DPP-4” erroneously indicated as “DDP-4”.

3)               Introduction, Line 55: “via [18GDG-] positron emission tomography (PET)”.

4)               Introduction, Line 70: empagliflozin repeated twice; canagliflozin missing.

5)               All through the manuscript: canagliflozin at 300mg is also a dual SGLT1 and SGLT2 inhibitor (not only sotagliflozin).

Author Response

We thank the reviewer for the insightful comments. We addressed all the raised issues, as follows.

 1)               Clinical evidence and epidemiological data come from observational studies, or from clinical trial not designed to verify the effect of SGLT2i on cancer. Therefore, many confounding factors could have influenced the results. First, the use of other antidiabetic drugs potentially influencing cancer incidence and outcomes (eg, metformin, secretagogues, insulin, GLP1Ras…). Also, different types of insulin used, mean daily doses of insulin, and glyco-metabolic control could play a role, and some other commonly used drugs could also interfere for these outcomes (eg. ASA and statins). Even if I personally agree with the pathophisiologically-based speculation of a positive effect of SGLT2i on cancer, these factors should be highlighted as potential limitations when describing epidemiological and clinical evidence.

We completely agree with the reviewer and added a paragraph at the end of the epidemiology section (highlighted in green, lines 321-326) taking into account the mentioned confounding factors. 

2) Since experimental and epidemiological data are conflicting and sometimes disappointing, as correctly admitted by the Authors, I think that some statements in the manuscript are exaggeratedly enthusiastic and should be attenuated

We did attenuate all the enthusiastic statements about the effectiveness of SGLT2 inhibitors in cancer (highlighted in green). Even if we are very excited about this potential therapeutic opportunity, we recognize that more evidence is required for their efficacy against cancer.

3)               The manuscript is very detailed, and overall could result difficult to read. I would suggest shortening some paragraphs (eg, “in vivo preclinical evidence) by taking advantage of the useful tables provided as supplements.

We significantly shortened the in vivo and in vitro preclinical evidence sections, removing some details that are presented in the tables.

Minor issues

 1)               SGLT2 inhibitors are sometimes indicated extensively, sometimes as SGLT2is. Please, unify. 

We replaced "SGLT2 inhibitors" with "SGLT2i" throughout the manuscript.

2)               Lines 204, 249, 252, 278: “DPP-4” erroneously indicated as “DDP-4”.

Sorry about that. We corrected the mistake

3)               Introduction, Line 55: “via [18GDG-] positron emission tomography (PET)”.

We added FDG to the positron emission tomography description.

4)               Introduction, Line 70: empagliflozin repeated twice; canagliflozin missing.

Sorry about that. We corrected the mistake

5)               All through the manuscript: canagliflozin at 300mg is also a dual SGLT1 and SGLT2 inhibitor (not only sotagliflozin).

Thank you for the specification. We corrected this (highlighted in green).

Round 2

Reviewer 1 Report

Comments and Suggestions for Authors

The authors performed my recommendations and I recommend publishing. The only item left is the quality of the figures. They must be improved.